# Experimental Study on the Effects of Vegetation on the Dissipation of Supersaturated Total Dissolved Gas in Flowing Water

**DOI:** 10.3390/ijerph16132256

**Published:** 2019-06-26

**Authors:** Zhenhua Wang, Jingying Lu, Youquan Yuan, Yinghan Huang, Jingjie Feng, Ran Li

**Affiliations:** 1State Key Laboratory of Hydraulics and Mountain River Engineering, Sichuan University, Chengdu 610065, China; 2Power China Zhongnan Engineering Corporation Limited, Changsha 410000, China

**Keywords:** supersaturated TDG, vegetation, retention time, dissipation coefficient, mitigation measures

## Abstract

High dam discharge can lead to total dissolved gas (TDG) supersaturation in the downstream river, and fish in the TDG-supersaturated flow can suffer from bubble disease and even die. Consequently, it is of great value to study the transport and dissipation characteristics of supersaturated dissolved gas for the protection of river fish. Floodplains may form downstream of high dams due to flood discharge, and the plants on these floodplains can affect both the hydraulic characteristics and TDG transport of the flowing water. In this study, the velocity distribution and the retention response time under different flow conditions and vegetation arrangements were studied in a series of experiments. The retention time was significantly extended by the presence of vegetation, and an empirical formula for calculating the retention time was proposed. In addition, the responses of the dissipation process of supersaturated TDG to hydraulic factors, retention time, and vegetation area coefficient were analyzed. The dissipation of supersaturated TDG significantly increased with increases in the vegetation area coefficient in the water. To quantitatively describe the TDG dissipation process in TDG-supersaturated flow under the effect of vegetation, the TDG dissipation coefficient was fitted and analyzed. The basic form of the formula for the dissipation coefficient involving various influence factors was determined by dimensional analysis. An equation for calculating the TDG dissipation coefficient of flowing water with vegetation was proposed by multivariate nonlinear fitting and was proven to have great prediction accuracy. The calculated method developed in this paper can be used to predict TDG dissipation in flowing water with vegetation and is of great significance for enriching TDG prediction systems.

## 1. Introduction

In recent years, China has made rapid progress in hydropower cascade exploitation, especially in the construction of high dams [1]. Along with the operation of super-high dams, such as Xiluodu, Xiaowan, and Jinping I, and the construction of dams, such as Wudongde and Baihetan, the impact of high dams and large reservoirs on the ecological environment has become increasingly prominent, and relevant research has also received extensive attention. High dams and large reservoirs play a role in flood regulation, storage, and process redistribution, which can bring into play important flood control benefits as well as change natural flood processes [2,3,4]. Floods play an extremely important role in river ecosystems [5]. Floodplains formed by flooding retain the material carried by the floods in the bottomland and therefore represent important places for fish to feed and spawn [6]. The flood peak process also provides spawning signals for fishes [7,8]. Flood transmission in a reservoir system occurs through flood discharge structures. In this process, the discharging jet entrains air, forming an aerated flow. As the water-air jet enters the deep region of a stilling basin due to the high head and high velocity, gas dissolves into the water under high pressure, and the total dissolved gas (TDG) content of the water reaches a state of supersaturation relative to the local atmospheric pressure [9,10]. The supersaturated gas dissolved in the water cannot dissipate rapidly, and these high TDG levels last for a considerable distance [11,12]. This phenomenon may cause fish in the lower reaches of the river to suffer from bubble disease or even death [13,14].

Many preliminary observations have shown that TDG supersaturation is common under dam discharge conditions. The problem of TDG supersaturation caused by dam discharge was first observed in the United States in the 1960s [15]. In China, the supersaturation of dissolved gas caused by discharge from the Gezhouba, Three Gorges, and Xiluodu dams has been reported to be a disaster for fish [16,17]. To mitigate the adverse impact of supersaturated TDG on the ecological environment, it is necessary to explore the transmission process of supersaturated water flow and its complex dissipation characteristics to promote the study of supersaturated gas mitigation measures.

Flood discharge from high dams causes the water level of the downstream reaches of rivers to rise and form floodplains in some areas. The existence of beach vegetation has a significant impact on the flow trajectory and hydrodynamic characteristics [18,19,20,21] and changes the transport and deposition processes of the water [22]. Initially, studies on flowing water with vegetation first identified the effect of vegetation on water flow resistance. W. O. Ree et al. estimated the retardation factor of a sink on the basis of vegetation density, vegetation height, and other related factors [23]. Shih et al. and Fathi-Maghadam et al. established an empirical relationship between vegetation resistance and water flow by considering the influence of vegetation density, characteristics, and distribution [24,25]. In terms of hydrodynamic characteristics, Yang studied the turbulent characteristics of overbank flow under the action of different floodplains with vegetation and found that the intensity of flow turbulence under the influence of different floodplains with vegetation was completely different [26]. Bai established a depth-averaged two-dimensional shallow water model to simulate the interaction between water flow and rigid vegetation [27]. Based on momentum transfer theory, Jiang established an analysis model for the flow velocity and flow distribution of submerged vegetation in a floodplain in a compound channel [28]. On the basis of predecessors, Huai et al. established an analytical model for predicting the vertical average velocity distribution in an open channel with double-layered rigid vegetation [29]. Han made use of a mixed-layer model by considering secondary flow, vegetation drag, and turbulent shear flows to more accurately predict the transverse velocity distribution in a channel with an uneven distribution of submerged vegetation [30]. On the basis of the above, some scholars have studied the transport process of substances in water under the action of vegetation. Liu found that planting flexible artificial grass on floodplains could significantly reduce the transport capacity of the whole channel [31]. Huai et al. examined the longitudinal dispersion coefficient under the condition that rigid emergent vegetation grows symmetrically along the riverbank and established a three-zone model for estimating the longitudinal diffusion coefficient [32]. Okamoto studied the mass transport and substance transport in an open channel by changing the density of rigid vegetation and continuously injecting dye [33]. Wu et al. introduced vegetation density into a two-dimensional model to simulate sediment transport in river and estuarine wetland areas and then studied the movement of sediment in water flow under high-density vegetation [34]. Considering that the existence of overbank vegetation changes the hydraulic characteristics and material transport process of water flow, Yuan studied the adsorption effect of vegetation on the dissipation of supersaturated TDG in static water containers with different vegetation densities and concluded that the wall adsorption coefficient was closely related to the material characteristics of the vegetation [35]. Yuan also initially proposed a three-dimensional two-phase flow transport and dissipation model for supersaturated TDG dissipation with vegetation based on OpenFoam [36]. However, there is still a lack of analysis on factors affecting the dissipation of supersaturated TDG in flowing water with vegetation and quantitative description methods for the impact of this process. This paper carried out a series of mechanism experiments. The comprehensive influence of vegetation wall area, average water depth, flow rate, average retention time, and other factors contained in water was analyzed. A relationship between the dissipation of supersaturated TDG and related factors under dynamic water conditions was established, and an expression of the dissipation coefficient of supersaturated TDG in flows with vegetation was also established. The research results will provide technical support and a reference for research on the effects of vegetation in water on the dissipation process of supersaturated TDG.

## 2. Experiment

### 2.1. Experimental System

This experiment was conducted at the State Key Laboratory of Hydraulics and Mountain River Engineering at Sichuan University, and the experimental system is shown in Figure 1. The flume channel is made of plexiglass and has a length of 1500 cm, a width of 50 cm, a height of 30 cm, and a slope of 0.45%. Plexiglass columns with a diameter of 1 cm and a height of 25 cm were used to simulate emergent vegetation in the flume. In this experiment, TDG-supersaturated flow was produced by a device to study the generation of TDG supersaturation in water and its effect on fish [37].

### 2.2. Measuring Methods and Instruments

In this experiment, the factors to be measured included TDG saturation, flow rate, velocity, average water depth, and average retention time of water. The water temperature was monitored during the experiment.

The TDG saturation and water temperature were measured by two total dissolved gas pressure (TGP) detectors positioned upstream and downstream of the flume. The flow rate was measured by a triangular thin-walled weir placed in the front of the flume, and the flow velocity was measured by a propeller current meter. The flow velocity in the flume was distributed in a symmetrical manner in the transverse direction, so the average flow velocity in the left part of the flow direction was measured as the overall value of the flow velocity. The average water depth was measured by a steel ruler. In each experimental condition, the average water depth of three sections in the upper, middle, and lower reaches of the intake flume was taken as the average water depth of the flume.

The average retention time represents the length of time required for the flow to pass through the upstream and downstream monitoring sections, and it was measured by the buoy method, which is described below.

Plastic tracers with a diameter of 1 cm were selected as the buoys. In each case, 20 tracers were put into the flow in the upstream section of the flume, and the length of time required for each tracer to pass through the upstream and downstream monitoring sections was measured. The process of each tracer passing through the flume was clearly recorded with a camera, and the retention time was calculated based on the video.

The TGP detectors used for the measurements were produced by Pentair Company of the United States. For these devices, the TDG saturation range is 0% to 200%, the precision is 2%, the temperature range is 0–40 °C, and the precision is 0.2 °C. The hydrometric propeller was an LGY-2 intelligent flow meter produced by Nanjing Shenglong Instrument Equipment Co., Ltd. of China. For this device, the measuring range is 1–300 cm·s^−1^, and the precision is 1 cm·s^−1^.

### 2.3. Experimental Conditions

Considering the water supply capacity of the experimental system and the geometric dimensions of the flume with vegetation columns, this experiment involved orthogonal conditions with 5 different vegetation densities and 5 water flow conditions. The flow rates were 1.5 L·s^−1^, 3.5 L·s^−1^, 5.5 L·s^−1^, 7.5 L·s^−1^, and 9.5 L·s^−1^. The vegetation density in the water is represented by the vegetation area coefficient *d_A_*, which is defined as the material wall area present in a unit volume of water. The vegetation area coefficient was set to 0, 0.90 m^−1^, 1.66 m^−1^, 2.54 m^−1^, and 4.71 m^−1^. The vegetation in the experimental section of the experimental flume was evenly arranged, as shown in Figure 2. The statistics for each experimental condition and result are shown in Table 1.

## 3. Results and Discussions

The following results were obtained for the flow velocity distribution, retention time, TDG dissipation, and supersaturated TDG dissipation coefficient under the influence of vegetation and the other experimental conditions.

### 3.1. Velocity

The velocity distribution in the flume has a great influence on the mass transport and substance transport, and this distribution is significantly altered by the presence of vegetation. Additionally, the average water retention time, which affects the dissipation process of supersaturated TDG, was also determined by the flow velocity.

The water flow direction was changed by the presence of vegetation. Additionally, complex flow patterns and velocity distributions were also formed by vegetation in vegetation flow areas. With increases in the vegetation area coefficient, the flow field characteristics under the influence of vegetation became more complex. Figure 3 shows the velocity distributions of different cross sections of the flume for a flow rate of 9.5 L·s^−1^ and vegetation area coefficients of 0.90 m^−1^ (case 25) and 4.71 m^−1^ (case 10), and the positions of the sections in Figure 3 can be determined according to Figure 2.

The cross-sectional velocity distributions of the column in cases 10 and 25 are shown in Figure 3a. The overall velocity distribution of the cross section of the column presents a sawtooth shape. The data for case 25 (with a flow rate of 9.5 L·s^−1^ and a vegetation area coefficient of 4.71 m^−1^) show that under the influence of vegetation, the cross-sectional velocity distribution of the column presents low velocities in the middle and high velocities on both sides, and the maximum velocity difference reaches 0.08 m·s^−1^. However, this phenomenon is not obvious in case 10 (with a flow rate of 9.5 L·s^−1^ and a vegetation area coefficient of 0.90 m^−1^). The results show that the higher the vegetation area coefficient in water is, the more obvious the influence on the velocity distribution.

The downstream cross-sectional velocity distributions of the column in cases 10 and 25 are shown in Figure 3b. Due to the blocking effect of the columns, the water flow formed a low-speed zone downstream of the columns. The velocity distribution in the cross section downstream of a vertical column is similar to that in the cross section passing through the vertical column; both feature a sawtooth distribution. In general, under the water-blocking effect of a column, the transverse velocity in the flume shows a trend with a low-velocity center and high-velocity edges. Compared to that in case 10 (with a flow rate of 9.5 L·s^−1^ and a vegetation area coefficient of 0.90 m^−1^), the vegetation area coefficient in case 25 (with a flow rate of 9.5 L·s^−1^ and a vegetation area coefficient of 4.71 m^−1^) is higher, the water-blocking effect is greater, and the above trend is more obvious.

### 3.2. Retention Time

Retention time is an important factor affecting the dissipation of supersaturated TDG [38], and this factor determines the time available for TDG adsorption on the vegetation surface and the wall surface of the flume. The average retention times in different cases were measured, and the correlations between the average retention time and various factors (Figure 4) were studied. The results show that the average retention time decreases with increasing flow rate, increases with increasing vegetation area coefficient, and increases with increasing average hydraulic radius. In the cases with a flowrate of 1.5 L·s^−1^, the maximum difference in the average retention time was 34% among different water vegetation area coefficients. Under the conditions with a vegetation area coefficient of 4.7 m^−1^, the average retention time of the water differed by 121% under different flow conditions. Thus, the average retention time under different conditions varied greatly. The influence of the flow on average retention time is greater than that of the vegetation area coefficient.

According to the above analysis, flow rate, vegetation area coefficient, and average hydraulic radius all have a certain influence on average retention time. To comprehensively understand the relationships between average retention time and the above three factors, the influence of each factor on average retention time was fitted by multiple nonlinear fitting, which is based on the analysis of the correlation between a single factor and average retention time. The results are as follows:(1)$$T=\alpha \times Q^{\beta }\times \exp(\chi d_{A})\times R^{\delta },$$ where *α*, *β*, *χ* and *δ* are dimensionless constants, and their values were obtained by mathematical fitting using MATLAB based on the experimental results for the 13 odd conditions in Table 1. By substituting each parameter value into Equation (1), the average retention time of water passing through the floodplain under the action of vegetation can be expressed as follows:(2)$$T=9.52\times Q^{-0.63}\times \exp(0.035d_{A})\times R^{0.44}.$$

Equations (1) and (2) are empirical formulas without real physical meaning, and the dimensions on both sides of the formula are not transferred to be the same. Equation (2) is used to calculate the average retention time *T_cal_* for the 12 even-numbered conditions in Table 1, and each value is compared with the average retention time *T_exp_* measured in the experiment to analyze the correlation between these variables, as shown in Figure 5.

According to Figure 5, *T_cal_* and *T_exp_* have a good correlation, and the Pearson correlation coefficient of these two variables is calculated to be 0.997, which indicates that Equation (2) has good applicability for calculating the average retention time of water with vegetation.

### 3.3. TDG Dissipation Process

Based on a previous study, the dissipation process of TDG in the flow with vegetation can be divided into three parts [35]: (1) Bubble-liquid mass transfer occurs due to decreasing pressure and turbulence in the water body, and this process can be affected by hydraulic parameters such as water depth and velocity. This process is the dominant factor in TDG dissipation. (2) Wall adsorption also causes TDG dissipation and is mainly influenced by the material properties of the solid walls. This process plays a significant role in TDG dissipation in flows with vegetation. (3) Air-water surface mass transfer also contributes to TDG dissipation and is mostly impacted by the wind speed and the TDG concentration in the top layer of the waterbody. In this experiment, there was no wind in the laboratory.

The relationships between TDG dissipation (TDG saturation difference at the inlet and outlet of the flume) and retention time, vegetation area coefficient, average water depth, and average flow velocity under various working conditions are shown in Figure 6.

According to Figure 6a, under the same flow rate, the relationship between the average retention time and the TDG dissipation is obvious. The longer the average retention time is, the greater the TDG dissipation is. For the conditions with a flow rate of 1.5 L·s^−1^, the TDG dissipation amount increased by 10.2% when the average retention time increased from 102 s to 137 s. The retention time determines the degree of the contact between the supersaturated water and the vegetation surface. The longer the average retention time is, the more gas will precipitate onto the vegetation surface, thus promoting the dissipation of supersaturated TDG.

As displayed in Figure 6b, when there were no columns in the flume, the TDG dissipation of the control group was 4.2% to 7.3%, while the TDG dissipation reached 7.0% to 17.5% under conditions with a vegetation area coefficient of 4.71 m^−1^. The difference was 2–3 times greater than that of the blank group. Therefore, vegetation has a promoting effect on the dissipation of supersaturated TDG. The greater the vegetation area coefficient is, the stronger the promoting effect is. This phenomenon becomes more significant with increasing flow. This phenomenon occurs because the larger the vegetation area coefficient is, the more complex the flow conditions are, and complex flow conditions promote TDG dissipation. In addition, high-density vegetation provides a larger surface area for bubble attachment, further promoting the dissipation of TDG.

According to Figure 6c, TDG dissipation increases with increasing average water depth under the same flow rate. When the flow rate was 1.5 L·s^−1^, the TDG dissipation increased from 7.3% at an average depth of 0.022 m to 17.5% at an average depth of 0.030 m. This result is not completely consistent with the previous research results of other scholars, who concluded that TDG dissipation increased with decreasing water depth in natural rivers. Because the average water depth in this experiment is small (for the same flow conditions, the largest difference in the average water depth is 0.03 m), this change has little effect on the dissipation of supersaturated TDG. Therefore, at the same flow rate, the effect of the vegetation area coefficient on the dissipation of supersaturated TDG is much greater than that of the average water depth, resulting in the phenomenon shown in Figure 6c.

Figure 6d shows an obvious relationship between the TDG dissipation and the average flow rate in this experiment. Specifically, the larger the average flow rate is, the smaller the TDG dissipation is. When the flow rate was 1.5 L·s^−1^, the TDG dissipation decreased from 17.5% at an average flow rate of 0.10 m·s^−1^ to 7.3% at an average flow rate of 0.14 m·s^−1^. At the same flow rate, the flow velocity is mainly determined by the vegetation area coefficient. The smaller the vegetation area coefficient, the higher the flow velocity is, and the water retention time also decreases. According to Figure 6a,b, decreases in both water retention time and vegetation area coefficient inhibit the dissipation of supersaturated TDG, resulting in the phenomenon shown in Figure 6d.

Based on the above experimental results, the presence of vegetation affects the dissipation of supersaturated TDG in many ways. For example, the presence of vegetation prolongs the average retention time, and the water is in full contact with the vegetation surface, which is conducive to the precipitation of gas. Additionally, the larger the vegetation area coefficient is, the larger the surface area for bubble attachment, and thus, more TDG will precipitate on the vegetation surface. Furthermore, under the influence of high-density vegetation, the hydraulic characteristics of water are strongly affected by the change in water flow. For example, both the average water depth and the water turbulence increase, thereby promoting the dissipation of TDG.

### 3.4. The Dissipation Coefficient of Supersaturated TDG

Researchers at the University of Washington proposed that “the dissipation process of supersaturated TDG in the downstream river follows the first-order kinetic reaction” [39,40,41]. The formula can be expressed as follows:(3)$$\frac{d(G-G_{\mathrm{eq}})}{dt}=-k_{T}(G-G_{\mathrm{eq}}),$$ where *G* is the TDG saturation at time *t* (%), *G*_0_ is the saturation at the initial moment (%), *G_eq_* is the equilibrium saturation of TDG (%), *t* is the retention time (s), and *k_T_* is the dissipation coefficient of TDG (s^−1^).

In Equation (3), the dissipation coefficient *k_T_* represents the rate at which supersaturated TDG dissipates and is an important indicator of the speed of supersaturated TDG dissipation. Based on Equation (3), the dissipation coefficient is calculated under various experimental conditions in this paper, as shown in Table 2, and the relation between the dissipation coefficient *k_T_* and the vegetation area coefficient is shown in Figure 7. When the flow rate is constant, the dissipation coefficient *k_T_* increases with increases in the vegetation area coefficient.

To further study the effect of vegetation on the dissipation of supersaturated TDG, the dissipation coefficient was further analyzed. Compared to a previous study, this study examined a higher average hydraulic radius and vegetation area coefficient in the analysis of the influential factors, and the expression of the dissipation coefficient was studied under the effects of various factors.

To obtain a final dissipation coefficient expression with actual physical meaning, the relationships between the dissipation coefficient and various physical quantities were analyzed by the Buckingham π theorem. Based on previous analyses, six physical quantities, *ρ*, *g*, *H*, *R*, *v*, and *d_A_*, were chosen for dimensional analysis, and *ρ*, *g*, and *H* were used as fundamental physical quantities. The form of the dissipation coefficient formula is as follows:(4)$$k_{T}=\sqrt{g/H}\times f(R/H,v/\sqrt{gH},d_{A}H),$$

In Equation (4), *f* represents the function composed of three dimensionless variables, R/H, $v/\sqrt{gH}$ and $d_{A}H$. The term $v/\sqrt{gH}$ represents the Froude number. In this formula, *f* can be regarded as a function of R/H, $F_{r}$ and $d_{A}H$. The values of these three dimensionless parameters in each case are shown in Table 3.

Based on the data from the 12 even-numbered conditions in Table 3, the correlation between each single variable and *f* was analyzed. On this basis, multiple nonlinear fitting was carried out on *f* using MATLAB, and the following expression of *f* was obtained:(5)$$f=1.1\times 10^{-3}\times \exp(-3.1R/H+0.65d_{A}H)\times F_{r}{}^{-0.43}$$

Equation (5) was verified by using the experimental results for the 13 odd conditions in Table 3. The value of *f_cal_* calculated using Equation (5) was compared with the value of *f_exp_* calculated from the experimental data. The results of this comparison are shown in Figure 8.

According to Figure 8, *f_cal_* and *f_exp_* have a good correlation, and the Pearson correlation coefficient is calculated to be 0.966. Equation (5) was then substituted into Equation (4), and the following expression of the dissipation coefficient *k_T_* was obtained:(6)$$k_{T}=3.96\times \sqrt{g/H}\times \exp(-3.1R/H+0.65d_{A}H)\times F_{r}{}^{-0.43},$$

Equation (6) can be used to calculate the dissipation coefficient of supersaturated TDG under the action of vegetation and to predict the dissipation process.

## 4. Conclusions

Based on the dissipation mechanism of supersaturated TDG, this paper carried out an experiment to study the effects of vegetation on the dissipation of supersaturated TDG in flowing water. By measuring and analyzing the variation in average water depth, average velocity, average retention time, and TDG saturation under the influence of different vegetation area coefficients and flow rates, the following conclusions were reached:
(1)The relationship between each parameter and the vegetation area coefficient in water was obtained. When the average flow velocity changes in response to changes in the vegetation area coefficient, the velocity distribution characteristics also change. Specifically, the horizontal distribution is characterized by low flow velocities in the center of the flume and high flow velocities along the edges, and the distribution characteristics of the columns show a sawtooth distribution. The average retention time is affected by the vegetation area coefficient, flow rate, hydraulic radius and other factors, and an empirical formula was proposed based on those factors.(2)The presence of vegetation in the water significantly promotes the dissipation of supersaturated TDG. The presence of vegetation prolongs the average retention time of water and increases the contact surface for absorption, thereby accelerating the dissipation of supersaturated TDG.(3)The dissipation coefficient *k_T_* is the most important parameter representing the supersaturated TDG dissipation rate. Using the Buckingham π theorem, a portion of the experimental data was used to determine the dissipation coefficient. The quantitative relationships of the vegetation area coefficient, average flow velocity, average water depth, and average hydraulic radius with *k_T_* are shown. Other experimental data are well verified, indicating that the calculation method for the dissipation coefficient proposed in this paper can be used to calculate the dissipation coefficient of supersaturated TDG under the influence of vegetation and to predict the dissipation process.

## Figures and Tables

**Figure 1 ijerph-16-02256-f001:**
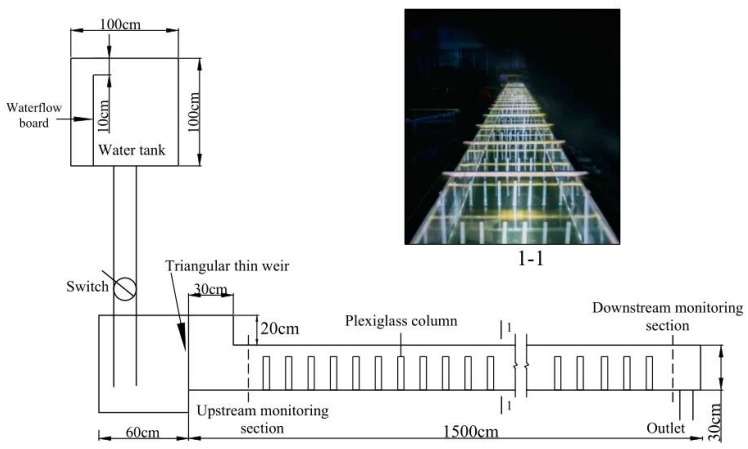
A sketch of the experimental flume.

**Figure 2 ijerph-16-02256-f002:**
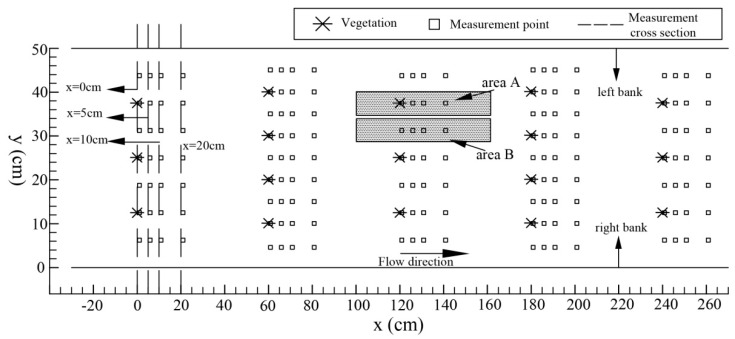
The arrangement of the glass columns in the flume (case 6 to case 10).

**Figure 3 ijerph-16-02256-f003:**
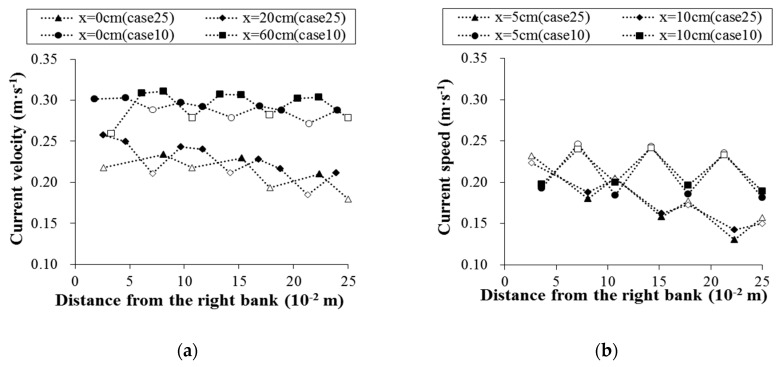
The velocity distribution of different cross sections (the solid and hollow symbols represent the velocities of areas A and B, respectively; the *x* axis represents the distance from the column): (**a**) The velocity distribution of the cross section of the column; (**b**) the velocity distribution of the downstream cross section of the column.

**Figure 4 ijerph-16-02256-f004:**
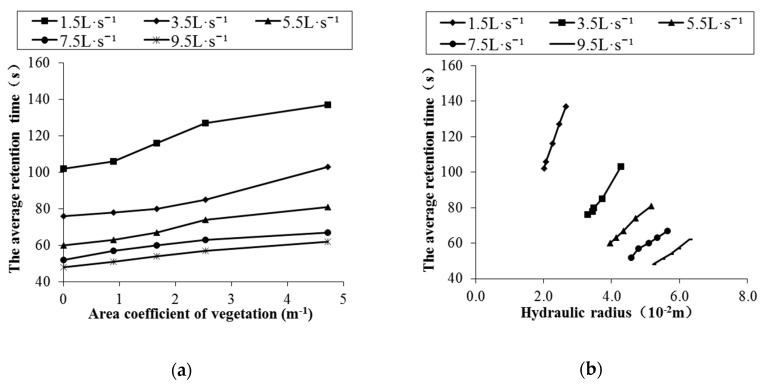
The effect of various factors on the average retention time under different flow rates: (**a**) area coefficient; (**b**) hydraulic radius.

**Figure 5 ijerph-16-02256-f005:**
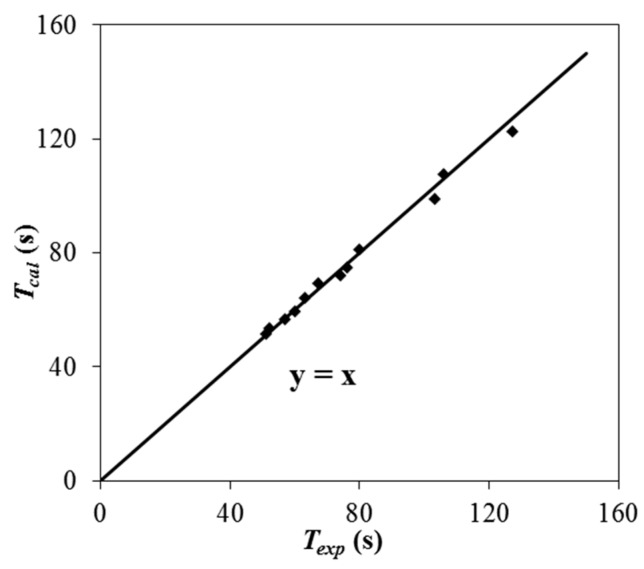
Comparison between *T_cal_* and *T_exp._*

**Figure 6 ijerph-16-02256-f006:**
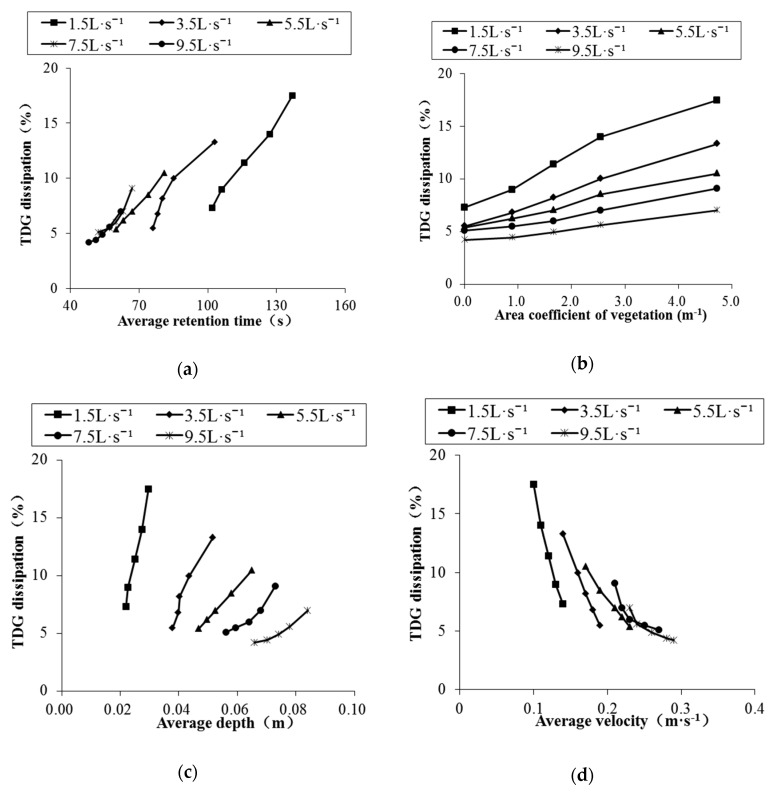
The relationship between TDG dissipation and various factors: (**a**) Average retention time; (**b**) area coefficient of vegetation; (**c**) mean water depth; (**d**) mean velocity.

**Figure 7 ijerph-16-02256-f007:**
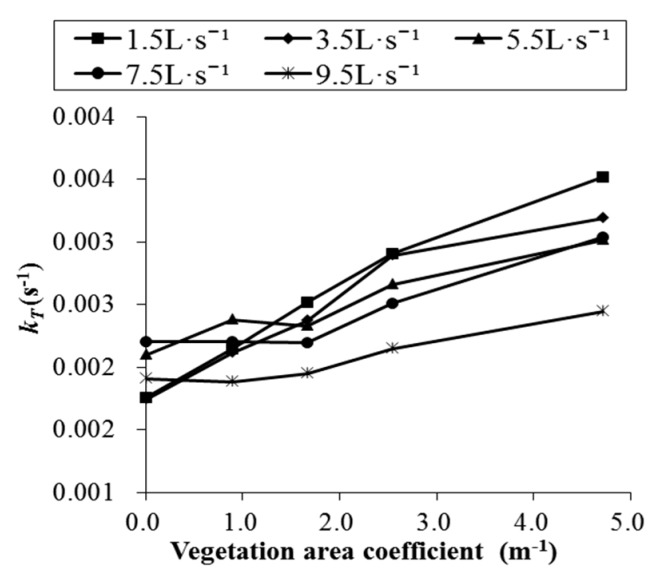
The relationship between *k_T_* and the vegetation area coefficient.

**Figure 8 ijerph-16-02256-f008:**
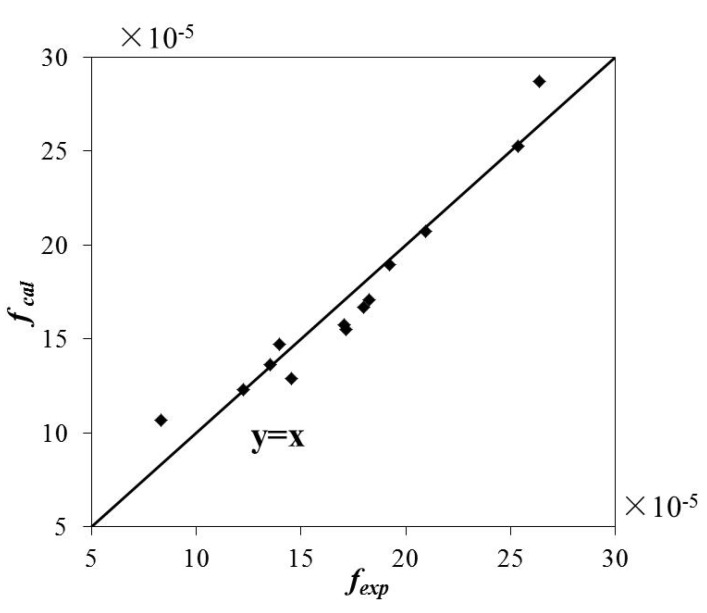
The relationship between *f_cal_* and *f_exp._*

**Table 1 ijerph-16-02256-t001:** The experimental cases and results.

Case No.	Flow Rate (m^3^·s^−1^)	Vegetation Area Coefficient (m^−1^)	Mean Depth (m)	Mean Hydraulic Radius (10^−2^ m)	Mean Retention Time (s)	TDG Dissipation (%)
1	0.0015	0	0.022	2.0	102	7.3
2	0.0035	0	0.038	3.3	76	5.5
3	0.0055	0	0.047	4.0	60	5.4
4	0.0075	0	0.056	4.6	52	5.1
5	0.0095	0	0.066	5.2	48	4.2
6	0.0015	0.90	0.023	2.1	106	14.0
7	0.0035	0.90	0.040	3.4	78	10.0
8	0.0055	0.90	0.050	4.1	63	8.5
9	0.0075	0.90	0.059	4.8	57	7.0
10	0.0095	0.90	0.070	5.5	51	5.6
11	0.0015	1.66	0.025	2.3	116	11.4
12	0.0035	1.66	0.040	3.5	80	8.2
13	0.0055	1.66	0.053	4.4	67	7.0
14	0.0075	1.66	0.064	5.1	60	6.0
15	0.0095	1.66	0.074	5.7	54	4.9
16	0.0015	2.54	0.027	2.5	127	14.0
14	0.0035	2.54	0.044	3.7	85	10.0
18	0.0055	2.54	0.058	4.7	74	8.5
19	0.0075	2.54	0.068	5.3	63	7.0
20	0.0095	2.54	0.078	5.9	57	5.6
21	0.0015	4.71	0.030	2.7	137	17.5
22	0.0035	4.71	0.052	4.3	103	13.3
23	0.0055	4.71	0.065	5.2	81	10.5
24	0.0075	4.71	0.073	5.7	67	9.1
25	0.0095	4.71	0.084	6.3	62	7.0

**Table 2 ijerph-16-02256-t002:** Total dissolved gas (TDG) dissipation coefficient under various conditions.

Case No.	Flow Rate (L·s^−1^)	Vegetation Area Coefficient (m^−1^)	Mean Depth (m)	Mean Hydraulic Radius (10^−2^ m)	TDG Dissipation Coefficient (10^−3^ s^−1^)
1	1.5	0	2.2	2.0	1.76
2	3.5	0	3.8	3.3	2.15
3	5.5	0	4.7	4.0	2.52
4	7.5	0	5.6	4.6	2.91
5	9.5	0	6.6	5.2	3.52
6	1.5	0.90	2.3	2.1	1.74
7	3.5	0.90	4.0	3.4	2.12
8	5.5	0.90	5.0	4.1	2.37
9	7.5	0.90	5.9	4.8	2.89
10	9.5	0.90	7.0	5.5	3.19
11	1.5	1.66	2.5	2.3	2.10
12	3.5	1.66	4.0	3.5	2.38
13	5.5	1.66	5.3	4.4	2.33
14	7.5	1.66	6.4	5.1	2.66
15	9.5	1.66	7.4	5.7	3.01
16	1.5	2.54	2.7	2.5	2.20
14	3.5	2.54	4.4	3.7	2.21
18	5.5	2.54	5.8	4.7	2.19
19	7.5	2.54	6.8	5.3	2.51
20	9.5	2.54	7.8	5.9	3.04
21	1.5	4.71	3.0	2.7	1.90
22	3.5	4.71	5.2	4.3	1.89
23	5.5	4.71	6.5	5.2	1.95
24	7.5	4.71	7.3	5.7	2.15
25	9.5	4.71	8.4	6.3	2.45

**Table 3 ijerph-16-02256-t003:** Values of the three dimensionless parameters

Case no.	Flow Rate (L·s^−1^)	Vegetation Area Coefficient (m^−1^)	*R/H*	*F_r_*	*d_A_H*
1	1.5	0	1.088	0.314	0.000
2	3.5	0	1.152	0.334	0.000
3	5.5	0	1.188	0.369	0.000
4	7.5	0	1.224	0.403	0.000
5	9.5	0	1.264	0.405	0.000
6	1.5	0.90	1.097	0.289	0.019
7	3.5	0.90	1.171	0.311	0.031
8	5.5	0.90	1.212	0.347	0.037
9	7.5	0.90	1.253	0.367	0.042
10	9.5	0.90	1.299	0.384	0.049
11	1.5	1.66	1.096	0.254	0.038
12	3.5	1.66	1.169	0.292	0.057
13	5.5	1.66	1.225	0.323	0.072
14	7.5	1.66	1.274	0.328	0.084
15	9.5	1.66	1.318	0.350	0.094
16	1.5	2.54	1.132	0.226	0.061
14	3.5	2.54	1.209	0.269	0.092
18	5.5	2.54	1.275	0.285	0.115
19	7.5	2.54	1.321	0.310	0.131
20	9.5	2.54	1.366	0.321	0.145
21	1.5	4.71	1.157	0.199	0.121
22	3.5	4.71	1.264	0.221	0.193
23	5.5	4.71	1.327	0.245	0.231
24	7.5	4.71	1.364	0.290	0.252
25	9.5	4.71	1.414	0.301	0.280

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
