# Peer review of "Experimental Study on the Effects of Vegetation on the Dissipation of Supersaturated Total Dissolved Gas in Flowing Water"

_ijerph, 2019, doi:10.3390/ijerph16132256_

Round 1

Reviewer 1 Report

This paper reports an investigation into the dissipation of total dissolved gas (TDG) supersaturation by submerged vegetation in a river or floodplain. Experiments were carried out in a flume in which equally spaced perspex columns served at models of trees or other vegetation. Parameters varied were flow rate, number of columns and hydraulic diameter of the columns. Total dissolved gas (TDG) was measured and its dependence on the parameters was studied.  A correlation was proposed based on dimensionless number analysis, and fitted coefficients and exponents determined.

The experimental measurements and statistical analysis seems to have been carried out competently, and the discussion makes good sense.  The paper is very well written, with clear explanation and arguments.

My only comment would be that I am not convinced that Froude number is actually the most physically relevant number to use in the correlation. While it would be certainly relevant to dam discharge situations, it is not clear to me what the relevance in this this situation: gravity seems to play no part, once the flowrate is set (apart from a minor influence on depth). Perhaps, in future, the authors could consider another characteristic velocity to non-dimensional the average flow velocity – perhaps something related to the velocity deficit caused by a column?

At any rate I have no hesitation recommending the paper for publication. I have marked a couple of minor suggestions related to the written expression on the manuscript attached.

Author Response

We are very grateful to your recognition about our paper, and your questions and suggestions are of grateful important for us. Now we want to give some explanations for your questions.

Point 1:

My only comment would be that I am not convinced that Froude number is actually the most physically relevant number to use in the correlation. While it would be certainly relevant to dam discharge situations, it is not clear to me what the relevance in this this situation: gravity seems to play no part, once the flowrate is set (apart from a minor influence on depth).

Response 1:

Thank you for your comment. With the effect of vegetation, it’s known that the flow regime is changed when compared to flow without vegetation, which can also impact on the TDG dissipation. In this paper, we use Froude number to present the effect to water flow regime. And the slope of the experimental water flume was 4.5‰, in which the gravity also played a role.

Point 2:

Perhaps, in future, the authors could consider another characteristic velocity to non-dimensional the average flow velocity – perhaps something related to the velocity deficit caused by a column?

Response 2:

Thanks a lot for your suggestion. We think the idea is amazing that a non-dimensional parameter is better to describe the effect of velocity variation on TDG dissipation. Also, our research group is working on the influence of velocity and turbulence on TDG dissipation, especially focusing on the region around vegetation columns. The original data are collected by ADV. We believe that the exploring of mechanism of the vegetation’s function of accelerating TDG dissipation will be developed in future.

Reviewer 2 Report

In this manuscript the authors perform some experiments to evaluate the influence of vegetation on the dissipation of supersaturated total dissolved gas (TDG) in a water channel. More in particular, they obtain an expression of the dissipation coefficient of supersaturated TDG.  This study is of interest for river fish protection and it is relevant for the field of this journal.

The paper is generally well written, anyway I have some comments and suggestions that could improve the manuscript:

1)     I think the Average Retention Time grows with the probability of tracer to be trapped into the cavity regions behind the Plexiglass columns. In your experiments you use plastic tracers with a diameter of 1 cm (comparable with the cavity region size), very bigger than size of real fluid particles. Don’t you think that could affect your results?

2)     It's not clear the meaning of x in Fig.3. You say it is the distance from the columns, but, if it was zero, it should correspond to a wall point with zero velocity. Further, could you draw the x axis coordinate in Fig. 2 ?

3)     In Equation (1), are T and Q dimensionless? If not, how can α and χ be dimensionless? At line 201 change Ď’ with χ.

4)     You could spend more words about the physical processes which affects the dissipation of TDG. I think it is not only the contact between supersaturated water and the vegetation surface, you cited.

5)     I think turbulence intensity could be very important in the phenomena you are studying, but you never show any measure of that.

6)     I don't agree with your phrase at lines 260-261 (maybe I didn't understand that), if you keep the flow rate constant, you can’t increase both average depth and flow velocity at the same time (due to mass balance).

7)     In table (2) you report Froude number before you define it.

8)     In Equation (4) dAH is not a dimensionless variable.

9)     Before to assert the general validity of your dissipation coefficient formula, you should test it by different experimental campaigns.

Author Response

Thanks for your recognition about our paper, and some of your suggestions are very helpful for our work. Here are reply about your questions and suggestions, and we deeply look forward to your further guidance.

Point 1:

I think the Average Retention Time grows with the probability of tracer to be trapped into the cavity regions behind the Plexiglass columns. In your experiments you use plastic tracers with a diameter of 1 cm (comparable with the cavity region size), very bigger than size of real fluid particles. Don’t you think that could affect your results?

Response 1:

We quite agree with your option that some of the tracers were trapped behind columns, even the tracers were there for quite a long time.

In the experiment, we employed 20 tracers in each measure group and there were two to three measured group in each case. The process about how each tracer go through the flume was clearly recorded by camera, and the retention time was calculated based on the video. Tracers that were trapped for a long time were abandoned. Then the rest tracers were used to calculate the Average Retention Time of each case.

By further reflection, we think your suggestion about the size of the tracers is very helpful for our further study. In future experiments we will use more reasonably sized tracers.

The above modification were marked yellow in line 124 to 131 in our new manuscript.

Point 2:

It's not clear the meaning of x in Fig.3. You say it is the distance from the columns, but, if it was zero, it should correspond to a wall point with zero velocity. Further, could you draw the x axis coordinate in Fig. 2?

Response 2:

Thanks for your reminding that it is not a clear illustration about the measured locations. We have modified the Fig.2 and added both x and y axis coordinate in Fig.2, furthermore, we marked the measured locations with square spots which can be compare with the vegetation column of circle spots. I hope the revising of Fig. 2 would help a lot understand the arrangement of the vegetation column and the representative of each curve in Fig.3.

Point 3:

In Equation (1), are T and Q dimensionless? If not, how can α and χ be dimensionless? At line 201 change ϒ with χ.

Response 3:

It was revealed that retention time was influenced by flow rate, area coefficient of vegetation and hydraulic depth with good correlations. Base on that, we concluded an empirical correlation as Equation (1) without real physical meaning. As a result, the dimensions on both sides of the formula are not transferred to be the same ,T and Q are dimensional. But in the future work, we will try to build a non-dimensional formula to calculate the Average Retention Time.

Point 4:

You could spend more words about the physical processes which affects the dissipation of TDG. I think it is not only the contact between supersaturated water and the vegetation surface, you cited.

Response 4:

Thanks for your suggestion. From prior work (Yuan, Y., Feng, J., Li, R., Huang, Y., Huang, J., & Wang, Z. Modelling the promotion effect of vegetation on the dissipation of supersaturated total dissolved gas. Ecological Modelling. 2018, 386, 89–97.), we consider that the dissipation of supersaturate TDG with vegetation could be divided into three parts, which are air-water surface mass transfer, bubble-liquid mass transfer and wall absorption, as shown in the following figure. In this paper, it was analysed that factors such as retention time, wall area of vegetation, velocity and depth could influence TDG dissipation process, which was illustrated in Fig. 6. Also as you comment that turbulence may also be an impact factor. But we didn’t measure the fluctuating velocity, the influence caused by turbulent characteristics was not included in this paper. We will accept reviewer’s suggestion to conduct analysis of effect of turbulence on TDG dissipation in next work.

In the discussing about impacts on TDG dissipation coefficient, retention time was removed from influence factors because the TDG dissipation coefficient was obtained by deriving time. It was analysed that the area of vegetation, flow regime are main factors influenced on TDG dissipation coefficient. In addition, the character of river cross-section, including contacting area of flow with wall and bed could also impact on TDG dissipation coefficient. Based on the analysis above, the calculated method of TDG dissipation coefficient in the flow with vegetation was discussed in this paper and then the equation (6) was proposed.

According to your comment, we have enriched the analysis of physical processes, in line 219-227 of our new manuscript?

Figure1. Three parts of the dissipation of supersaturate TDG in flow with vegetation

Point 5:

I think turbulence intensity could be very important in the phenomena you are studying, but you never show any measure of that.

Response 5:

We totally agree with you. The impact of turbulence was not discussed in this paper because fluctuation velocity was not measured in experiment.

We have gradually realized that turbulence could be an important factor. A member in our group is researching the influence of turbulence even micro eddy on TDG dissipation in the flow with vegetation, and her work will be published in another journal paper. In this paper, we discussed the hydrodynamic effects involving velocity and Froude number.

Point 6:

I don't agree with your phrase at lines 260-261 (maybe I didn't understand that), if you keep the flow rate constant, you can’t increase both average depth and flow velocity at the same time (due to mass balance).

Response 6:

Thanks for pointing out our error, there were some problems in the expression in this sentence, and we have already amended it. Please check it in line 271-273.

Point 7:

In table (2) you report Froude number before you define it.

Response 7:

Thanks for your suggestion. We did not mention Fr before table (2), and it appears in the results of dimensional analysis. Thus we delete the column represented Fr in the table (2), and make table (3) to put forward the value of  and .  in each case. Now we think our paper’s data looks more logical and clear owing to your suggestion.

Point 8:

In Equation (4) dAH is not a dimensionless variable.

Response 8:

Maybe we didn't make dA clearly. It is defined as the material wall area contained in a unit volume of water, and the unit of it is m-1, thus dAH is a dimensionless variable. To describe it more clearly, we have changed the vegetation area fraction into the vegetation area coefficient, and we redefine it in line 141-143.

Point 9:

Before to assert the general validity of your dissipation coefficient formula, you should test it by different experimental campaigns.

Response 9:

When fitting the formula of dissipation coefficient, as shown in Equation (6), 12 cases’ data in 25 experimental cases were selected. And the remaining 13 cases’ data were used to verify the established formula, as shown in figure 8. So the formula was built according to part of experimental data and verified by different data series. Also, we are looking forward to verifying this formula with other data from different research group or from our further exploring. We have revised line 297-310 to make it clearly to describe our steps of how to get the formula of Equation (6).

Reviewer 3 Report

The information presented in this manuscript is potentially useful to readers, but the presentation makes it extremely difficult to extract useful conclusions from the presentation.My main concern is that the interpretation of results is not very mechanistically based, making it very difficult to make much sense of what has been presented.  This has to be addressed at the very minimum.  It's frustrating when the communication is hampered by lack of facility with the English language but there are more important issues here. 

Based on my knowledge of the problem, the "dissipation" of TDG could be due to two influences, gas exchange across the water surface and a mechanism for which no physical explanation is provided but which seems to be tied to reference 35 which suggests that the vegetative surface impacts the loss of TDG although by what mechanism is totally not discussed.  I didn't try to look up that reference but I am hoping that some different types of material surfaces that somehow represent real vegetation as opposed to plexiglass were investigated.   It is not at all clear to me how plexiglass surfaces should have a major influence on TDG dissipation and I would like to see a "control" experiment that demonstrates that this is an important effect.  It is clear that the presence of vertical cylinders can impact the turbulence in the flow and therefore indirectly the surface gas transfer, somewhat consistent with other description of the gas dissipation process, but we should have of the relative contribution of the two effects.  This is especially critical since the experimental setup uses plexiglass rods that have a diameter of 1/50 of the channel width which seems large relative to what one would think that conventional vegetation would be. 

Some general comments on the presentation.  Area fraction as defined is inconsistent with the name which suggests a dimensionless quantity.  At least it is defined which is more than with some other experimental variables.  "retention time" is never really rigorously defined and there seems to be something wrong in any case.  A retention time for a plug flow system is defined as a volume divided by a discharge.  I can imagine what these variables would be in this experiment and therefore the retention time should go down with increasing area fraction (although the effect might be minor for low vegetation concentrations) because of the reduced volumes of the flow section, but the opposite effect is shown.  Since I don't know exactly how the retention time is actually calculated (never explained) I can't make any specific comment other than there is a problem here.  Equation 3 which is indicated to represent a first order process is not consistent with that representation and is illogical as presented.  This would need to be fixed if it is not simply an error in the manuscript.  Finally the representation of the gas transfer is not very consistent with general results in the literature.  The dimensional analysis is illogical in the sense that the gas transfer is given for the experimental system and therefore for a given length of channel and if the transfer is dependent on a time for gas transfer at the surface would be more appropriately defined in terms of the discharge rather than variables V and H which are related to it.  Thus retention time,  if properly determined) should be a more relevant variable.  This may be an issue with the use of plastic floats, but simply impossible t understand with the information presented.  There is a lot of work needed here and the presentation in the manuscript is naive.

The introduction and abstract discuss a floodplain, which to me implies a main channel and overbank flows in rougher and shallow sections.  But the experiments were performed so far as I can tell by a uniform distribution of vegetation elements and depth across the cross section.  So the situation they describe is not well represented in the experiment.

Measurements of velocity presented in Figure 3.  Since I don't know where the measurements were made relative to the vegetation elements, it's hard to know how to interpret.  What means x=0?  why are results presented at inconsistent locations?

Dissipation should not be dependent on the geometry of the specific experimental system in the case that somebody wanted to compare results from other similar experiments in the future.  There needs to be a better interpretation of results.

I have a whole list of comments that result from undefined quantities, lack of clarity in the presentation, or other grammar issues.  To keep from having to repeat all these now, if the authors were to choose to revise the manuscript and address the above issues, then I can weigh in on these other problems once the fundamental problems are addressed.

Author Response

Thanks for your recognition about our paper, and some of your suggestions are very helpful for our work. Here are reply about your questions and suggestions, and we deeply look forward to your further guidance.

Point 1:

Based on my knowledge of the problem, the "dissipation" of TDG could be due to two influences, gas exchange across the water surface and a mechanism for which no physical explanation is provided but which seems to be tied to reference 35 which suggests that the vegetative surface impacts the loss of TDG although by what mechanism is totally not discussed.  I didn't try to look up that reference but I am hoping that some different types of material surfaces that somehow represent real vegetation as opposed to plexiglass were investigated. It is not at all clear to me how plexiglass surfaces should have a major influence on TDG dissipation and I would like to see a "control" experiment that demonstrates that this is an important effect.  It is clear that the presence of vertical cylinders can impact the turbulence in the flow and therefore indirectly the surface gas transfer, somewhat consistent with other description of the gas dissipation process, but we should have of the relative contribution of the two effects. This is especially critical since the experimental setup uses plexiglass rods that have a diameter of 1/50 of the channel width which seems large relative to what one would think that conventional vegetation would be. 

Response 1:

Thanks for your professional question. Based on our team's years of work about TDG, we proposed a model that the dissipation of supersaturate TDG can divided into three parts, included air-water surface mass transfer, bubble-liquid mass transfer and wall absorption, as shown in Fig.1. Based on our prior research, the bubble-liquid transfer due to pressure variation is a dominate function of TDG dissipation. However, the wall absorption also plays an important role in the TDG dissipations process with vegetation, which is our concern in this paper.

Fig. 1 Sketch of the dissipation of supersaturated TDG

In reference 35 we conduct some experiments to research the wall absorption’s effect on the dissipation of supersaturated TDG. Artificial plants made of PVC and Plexiglas sheets made of PMMA were used as vegetation in the experimental. Five vegetation densities was designed including a control group without vegetation (Case 1) as shown in Fig. 2. The experimental results are illustrated in Fig. 3 that planting vegetation in water can effectively promote the dissipation of supersaturated TDG. The promoting effect of artificial plants (PVC) with rougher walls is superior to that of plexiglass sheets (PMMA) with smoother walls during the dissipation process of supersaturated TDG. So it was demonstrated that wall did accelerate TDG dissipation even if the surface was as smooth as plexiglass. Our group is working on what kind of material can promote TDG dissipation most and which characteristics are significant factor to influence the dissipation rate.

Fig. 2 Sketch of the experimental devices

Fig. 3 The TDG dissipation process of experimental cases

As to the columns’ diameter of 25px is up to a size 1/50 of the whole channel’s width, which may be not reasonable compared to real floodplain. Our explanation would be that it is possible that the arrangement of columns is more intensive than natural condition, but a diameter of a stem with 25px is normal. The aim we carry out experiments is to know mechanism or law of how the vegetation can promote the TDG dissipation. In the experiment design, there is also some objective factors that we need to concern. For example, there is only one reliable TDG measurement device in the world with accuracy of ±2%, which is Point Four Tracker made by Pentair, and the dissipated TDG saturation throughout a channel is often not a big value even if we had tried our best to build a channel with length of 15m in the experimental room. In this situation we had to enhance the vegetation density a little to obtain a more significant decreasing of TDG saturation to mitigate system error caused by measurement. In our opinion, the purpose of our research is to make clear how vegetation accelerate TDG dissipation in the flow, so we don’t need to copy a filed case with a scale. Our ultimate goal is to apply what we research into practice on a real floodplain, so thank you a lot for reminding us the problem.

Point 2:

Area fraction as defined is inconsistent with the name which suggests a dimensionless quantity. At least it is defined which is more than with some other experimental variables.

Response 2:

Maybe we didn't make area fraction clear. It is defined as the material wall area contained in a unit volume of water, and the unit of it is m-1, and dAH is a dimensionless variable. In the revision, we use “vegetation area coefficient” to replace “vegetation area fraction”, this content can be seen in line 141-143 in our new manuscript.

Point 3:

"Retention time" is never really rigorously defined and there seems to be something wrong in any case. A retention time for a plug flow system is defined as a volume divided by a discharge.  I can imagine what these variables would be in this experiment and therefore the retention time should go down with increasing area fraction (although the effect might be minor for low vegetation concentrations) because of the reduced volumes of the flow section, but the opposite effect is shown.  Since I don't know exactly how the retention time is actually calculated (never explained) I can't make any specific comment other than there is a problem here.

Response 3:

We have made a clear definition about the average retention time, it represent the time of flow through the upstream and downstream monitoring sections.

The average retention time was measured by the buoy method: Plastic tracers with a diameter of 1 cm were selected for the buoy. In each case, we put twenty tracers into the flow in the upstream section of the flume, the time of each tracer through the upstream and downstream monitoring sections was measured. The process about how each tracer go through the flume was clearly recorded by camera, and the retention time was calculated based on the video. Tracers that were trapped for a long time were abandoned, and the rest tracers were used to calculate the average retention time of each case.

The above content can be find in the manuscript in line 124-131.

Point 4:

Equation 3 which is indicated to represent a first order process is not consistent with that representation and is illogical as presented. This would need to be fixed if it is not simply an error in the manuscript.

Response 4:

Thanks for your indication about our mistake, equation 3 is indeed a wrong expression of the first-order kinetic reaction. We have modified it in the revision in line 277-278, and we feel very sorry to make this mistake.

Point 5:

Finally the representation of the gas transfer is not very consistent with general results in the literature. The dimensional analysis is illogical in the sense that the gas transfer is given for the experimental system and therefore for a given length of channel and if the transfer is dependent on a time for gas transfer at the surface would be more appropriately defined in terms of the discharge rather than variables V and H which are related to it. Thus retention time, if properly determined) should be a more relevant variable. This may be an issue with the use of plastic floats, but simply impossible understand with the information presented.  There is a lot of work needed here and the presentation in the manuscript is naive.

Response 5:

As we mentioned above that the dissipation progress of TDG in the flow with vegetation can be divided into three parts: (1) Bubble-liquid transfer due to pressure decreasing and turbulence in water body, which can be affected by hydraulic parameters such as water depth and velocity. (2) Absorption happen on solid surface also causes TDG dissipation, which is mainly influenced by material properties of solid walls. (3) Gas transfer at free water surface is another part of TDG’s dissipation, which is mostly impacted by the wind speed and the concentration of TDG in the top layer of waterbody. We consider that gas transfer at free water surface can be ignored since there is no wind in the laboratory room. In our experimental analysis, we believe that water depth and velocity is important factors for the dissipation of TDG. In addition, the interfacial area of vegetation is still an important factor affecting dissipation. Moreover, the flow pattern also has an impact on the dissipate rate, so Fr is selected as one of the influencing factors. Actually, as mentioned by other reviewer that turbulence intensity may be another significant factor, and we try to explore it in future to discuss the contribution of turbulence and vortexes to TDG dissipation in a microscopic perspective.

In the discussing about impacts on TDG dissipation coefficient, retention time was removed from influence factors because the TDG dissipation coefficient was obtained by deriving time. It was analysed that the area of vegetation, flow regime are main factors influenced on TDG dissipation coefficient. In addition, the character of river cross-section, including contacting area of flow with wall and bed could also impact on TDG dissipation coefficient. Based on the analysis above, the calculated method of TDG dissipation coefficient in the flow with vegetation was discussed in this paper and then the equation (6) was proposed.

Point 6:

The introduction and abstract discuss a floodplain, which to me implies a main channel and overbank flows in rougher and shallow sections. But the experiments were performed so far as I can tell by a uniform distribution of vegetation elements and depth across the cross section. So the situation they describe is not well represented in the experiment.

Response 6:

It is such a great question. We think it is not clear in the Introduction part and we have done some revision. The background of our study on this issue is that flood discharge from high dams always causes TDG supersaturation, which has a strong negative impact on fish. Whereas, along with cascade high dams building and operating in China, discharging from a dam’s flood spilling structure is the only way to pass the flood to meet the demands of aquatic organisms or flood control. At the same time, flood floodplain is an important habitat to feed fish with great rich bait in flood season. Considering the TDG supersaturation is impossible to avoid when a dam is discharging and flood plain is such an important region for fish, is there anything we can do to reduce a partial region’s TDG saturation by utilizing vegetation to decrease TDG level and mitigate harm to fish?Thus, we started our research to discover the mechanism of promoting TDG dissipation with vegetation.  It is sure that the riverbed structure of flood plain is compound channel. At the initial stage, we hope to first understand the influence of vegetation on the dissipation of supersaturated TDG in the flow. The experiment in this paper is focusing on the influence in the region of floodplain, so we planted vegetation in the whole lateral direction. In the future, we will expand the research to compound channel, involving the 3D distribution of hydrodynamics and TDG and exchange of momentum and energy between main channel and floodplain, which is carrying out by a Ph.D candidate.

Point 7:

Measurements of velocity presented in Figure 3. Since I don't know where the measurements were made relative to the vegetation elements, it's hard to know how to interpret. What means x=0?  why are results presented at inconsistent locations?

Response 7:

Thanks for your suggestion. Sorry for confusing you in the figure. We have modified the Fig.2 and added both x and y axis coordinate in Fig.2, furthermore, we marked the measured locations with square spots which can be compare with the vegetation column of circle spots. I hope the revising of Fig. 2 would help a lot understand the arrangement of the vegetation column and the representative of each curve in Fig.3.

Point 8:

Dissipation should not be dependent on the geometry of the specific experimental system in the case that somebody wanted to compare results from other similar experiments in the future. There needs to be a better interpretation of results.

Response 8:

The initial goal of our work would be exploring the dissipation process of TDG in the flow with vegetation and obtaining some useful results which is valid and can be use in other experimental or field studies. So we are a little confusing about your comment of “Dissipation should not be dependent on the geometry of the specific experimental system”. It is sure there is still a lot to be improved in the organization and expression of our research results this paper. We have tried to improve it to be more logical and we have sent it to an editing company to make the expression close to native speakers. We hope you can see the effort form our revision and reply to your professional comments. Thank you again for your guidance and suggestions.

Round 2

Reviewer 2 Report

I think the authors have addressed all the concerns in a correct way, with the exception of point 3.

It is not correct to write an equation in which left and right sides have different dimensions, you should at least include in the paper the answer you wrote in your comments of point 3. Furthermore (as I already said)  at line 207 change Ď’ to χ, this last is the variable you use in the equation 1.

Author Response

Point 1:

It is not correct to write an equation in which left and right sides have different dimensions, you should at least include in the paper the answer you wrote in your comments of point 3. Furthermore (as I already said) at line 207 change ϒ to χ, this last is the variable you use in the equation 1.

Response 1:

Thanks for put forward this question to us. Firstly, we changed ϒ to χ, and we indeed didn’t find this error before. Then, as we mentioned before, equation (1) and (2) are empirical formula. Based on the experimental data, the equations were carried out by multiple nonlinear fitting using MATLAB, so the dimensions on both sides of the formula are not transferred to be the same.

We took your advice, added the derivation process of the formula and explained why equation (1) and (2) have different dimensions in left and right sides in line 207 to 209, 211 to 212.

In our following work, we will try different methods in order to find a non-dimensional formula to accurately calculate the Average Retention Time.
